# Effect of Hen Genotype and Laying Time on Egg Quality and Albumen Lysozyme Content and Activity

**DOI:** 10.3390/ani13101611

**Published:** 2023-05-11

**Authors:** Józefa Krawczyk, Lidia Lewko, Zofia Sokołowicz, Anna Koseniuk, Adam Kraus

**Affiliations:** 1Department of Poultry Breeding, National Research Institute of Animal Production, Krakowska Street 1, 32-083 Krakow, Poland; 2Department of Animal Production and Poultry Products Evaluation, University of Rzeszów, Zelwerowicza 4, 35-601 Rzeszów, Poland; 3Department of Animal Molecular Biology, National Research Institute of Animal Production, Krakowska Street 1, 32-083 Krakow, Poland; 4Department of Animal Science, Faculty of Agrobiology, Czech University of Life Sciences Prague, Food and Natural Resources, Kamýcká 129, 165 00 Prague, Czech Republic

**Keywords:** egg quality, egg laying time, lysozyme, native breeds of hens

## Abstract

**Simple Summary:**

We studied the putative effect of egg laying time and genotype of selected Polish native breeds of hens on egg quality and lysozyme content and activity in the albumen. We collected eggs form hen breeds included in the conservation programme in Poland, namely: Green-legged Partridge (Z-11), Yellow-legged Partridge (Ż-33), Rhode Island Red (R-11) and Leghorn (H-22). We found significant differences in the lysozyme content and activity in thick and thin albumen in eggs laid by the studied hens, while no effect of egg collection time on this trait was noted.

**Abstract:**

According to research, egg quality and lysozyme content are influenced by a number of factors, which are mostly known in the case of commercial hybrids, while in breeds included in genetic resources conservation programmes, new research results in this regard are emerging. The aim of the study was to determine the effect of egg laying time and genotype of selected Polish native breeds of hens on egg quality and lysozyme content and activity in the albumen. The study material consisted of eggs collected from four strains of laying hens included in the Polish conservation programme, i.e., Green-legged Partridge (Z-11), Yellow-legged Partridge (Ż-33), Rhode Island Red (R-11) and Leghorn (H-22). At week 56, 28 eggs were randomly collected at 7:00 and 13:00 h from each breed of hen and assessed for quality. Laying time influenced certain egg quality traits. Eggs laid by hens in the morning were characterised by 1.7 g lower total weight and albumen weight, 2.4 pores/cm^2^ higher number of shell pores, 0.15 higher albumen pH values and 0.17 lower yolk pH values compared to those laid in the morning. The time of laying did not affect the level and activity of lysozyme in the albumen. A significant negative correlation was found between eggshell traits and albumen height, and between Haugh unit and lysozyme content and activity in the albumen. The studied egg quality traits were more influenced by genotype than by the egg-laying time.

## 1. Introduction

According to research, egg quality is influenced by a number of factors, which are mostly known in the case of commercial hybrids, while in breeds included in genetic resources conservation programmes, new research results in this regard are emerging [1,2,3,4,5,6,7,8]. In commercial flocks, egg weight not only determines the quantity of the basic components of the egg but also has a significant impact on the interior egg and shell quality; however, in populations of native breed hens, these relationships are not always similar. Egg weight depends on many factors, but most significantly on the genotype and age of the hens [9]. Approximately 60% of egg weight is albumen, which is a valuable nutrient for humans as it contains 11% protein, 0.6% carbohydrates, 0.4% minerals and additionally, many biologically active substances, including lysozyme. The lysozyme in egg albumen provides natural protection against infection of the embryo until the embryo produces immunoglobulins [10]. The chemical composition and quality of eggs as well as the level of lysozyme in albumen are influenced by the housing system of the hens [11]. Due to its antimicrobial activity, lysozyme is used in the pharmaceutical and food industries. The work in [10] also showed a positive effect of lysozyme on the hatchability of Japanese quails. The study in [6] demonstrates that in eggs from hen breeds included in genetic resource conservation programmes, lysozyme levels increase with the age of the hens. The authors of [12] also found an increase in lysozyme content in eggs with the age of Ross 308 hens, with the authors suggesting that this is due to a deterioration in shell quality in older hens. Refs. [2,13] noted, in populations of native and hens for backyard flocks, the presence of significant correlations between some egg quality traits and the concentration of lysozyme and ovotransferrin in the eggs, suggesting the possibility of using the eggs of these hens for lysozyme extraction.

Some studies conducted with commercial hybrid hens show that egg weight and shell quality vary according to the time of laying, the genotype and age of the hens [1,14]. Ref. [15] observed that layers with high laying capacity lay the most eggs early in the morning, and layers with lower productivity lay much later in the day. The time of laying has the greatest effect on egg weight, yolk percentage and shell strength. A study by [8] shows that egg weight influences most quality traits, while the crushing strength and shell colour of eggs from native breeds are not as uniform as among eggs from commercial hybrids. A study by [10] showed that shell colour can be a determinant of the antimicrobial potential of eggs, which was also confirmed by [5,16] for hens of native breeds. Refs. [17,18] pointed to the differences in the antimicrobial potential of the commercial lines of birds, which resulted from genetic work in breeds subjected to artificial selection on pedigree farms, considering the biological value of hatching eggs.

In our study, we used eggs collected from hens of four strains: Green-legged Partridge (Z-11), Yellow-legged Partridge (Ż-33), Rhode Island Red (R-11) and Leghorn (H-22). The Z-11 and Ż-33 breeds are old, native populations of hens, while the R-11 and H-22 are locally adapted breeds brought to Poland for breeding farms after World War II, but withdrawn from intensive production in the 1970s.

Regarding the studies conducted elsewhere, we formulated the following hypothesis: assuming that the laying time influences the quality of the eggs’ physical features, this factor may also affect the lysozyme content in the albumen. In breeding practices, eggs with higher content and enzymatic activity of lysozyme could be preferred.

The aim of the study was to determine the influence of egg laying time and genotype of selected hen breeds included in genetic resources conservation programmes on egg quality and the lysozyme content and activity in the albumen.

## 2. Materials and Methods

The research material consisted of eggs collected from 4 breeds/strains of laying hens included in the conservation programme in Poland, namely: Green-legged Partridge (Z-11), Yellow-legged Partridge (Ż-33), Rhode Island Red (R-11) and Leghorn (H-22). The birds were kept on the experimental farm, each strain in 4 replicates of 30 hens each, i.e., a total of 480 hens were included in the study. The hens were kept in a litter system at a stocking rate of 5 hens/m^2^ and fed a complete feed mixture recommended for laying hens, with free access to water and feed (ad libitum). The feed mixture contained: 92.31% dry matter, 11.63% crude ash, 15.33% crude protein, 3.09% crude fat and 3.15% crude fibre. The energy value of the feed mixture was 11.2 MJ/kg.

In the 56th week of life, hen laying was as follows: Z-11–63.1%; Ż-33–41.0%; R-11–44.2%; H-22–52.2%. Additionally, in the 56th week of hens’ age, 28 eggs were collected at 7:00 a.m. (7:00 h) and 1 p.m. (13:00 h) (a total of 224 eggs) from each breed of hens. The eggs collected at 7 a.m. were laid between 1 p.m. of the day before and 7 a.m. on the day of collection. The eggs collected at 1 p.m. were laid between 7 a.m. and 1 p.m. on the day of collection. Eggs were selected randomly, having previously discarded dirty eggs and eggs with cracked shells. Next, the eggs were subjected to quality assessment using the electronic EQM (Egg Quality Measurements) apparatus from TSS QCS-II, which measures albumen and Haught unit (HU). The number of pores per cm^2^ of shell (shell porosity) was determined according to [19]. The shell area on which the pores were counted was 0.25 cm^2^. Shell crushing strength (N) was tested with a texture analyser TA.XT PLUS (Stable Micro Systems, Godalming, UK/2010) fitted with appropriate attachments. The shell thickness (µm) was measured after removing shell membranes using a Mitutoyo Digital Micrometer 293-766-30 (Mitutoyo America Corporation, Aurora, IL, USA). The colour of the yolk and shell (L*, a*, b* colour space) was measured using the Konica Minolta CM-700d spectrophotometer (Osaka, Japan, 2018). The instrument was calibrated with a white reference plate (Konica Minolta, Sensing, Inc., Japan), with setting values (L* = 97.10, a* = −4.88, b* = 7.04) before the measurement. Results were recorded as L*, a*, b* colour space. The L* value represents lightness (0 = black, 100 = white), a* indicates redness (−100 = green, 100 = red) and b* gives a value for yellowness (−100 = blue, 100 = yellow)**.** The pH measurement of albumen and yolk was performed using a Seven2Go pH meter (Mettler-Toledo, Urdorf, Switzerland). The measurement was carried out by immersing the electrode in the particular fractions of the egg.

To conduct a detailed analysis of the active ingredient (lysozyme) in egg albumen fractions (hydrolytic activity, percentage), the previously collected and weighed eggs from different experimental groups of the birds were broken and separated into albumen and yolk. Next, the albumen was separated into thick and thin fractions and placed in disposable sterile containers. The so-prepared albumen samples were used to make preparations to determine the concentration and hydrolytic activity of lysozyme in the albumen fractions. The content and activity of lysozyme in the fresh albumen fractions was determined with spectrophotometry using a spectrophotometer (Metertech SP-830 plus; Nangang, Taipei, Taiwan/2011) [20], consisting in the use of the lysis of cell walls of Micrococcus lysodeikticus bacteria. Hydrolytic activity of lysozyme was expressed in lysozyme activity units (U/mL). One such unit is defined as the amount of lysozyme that in one minute decreases the absorbance of a Micrococcus lysodeikticus bacterial suspension by 0.001, measured at a wavelength of 450 nm and a temperature of 25 °C. The reaction occurred in a mixture of a volume of 2.6 mL (2.5 mL of bacterial suspension +0.1 mL lysozyme solution) and pH 6.24, in a cuvette with a length of optical path being 1 cm. After calculating the value of absorbance decrease (Δ*A*) for the working solution of lysozyme, a curve of correlations between Δ*A* and lysozyme concentration was graphed. Next, based on the standard curve, hydrolytic activity of lysozyme was determined in the studied preparations. 

The value of the decrease in the solution absorbance (Δ*A*) (U/min.) was calculated from the following equation: ΔA=At0−At

*At*0—absorbance value for bacterial suspension at time *t*0;

*At*—absorbance value for bacterial suspension after time *t*.

Data were processed using Statistica 13.1 PL software (StatSoft Polska Sp. z o.o., Krakow, Poland). Mean values for all analysed parameters were calculated. A two-way model of ANOVA was used to analyse variability (variable 1: time of oviposition; variable 2: genotype). The significance of differences was verified using Duncan’s test. Interactions between experimental variables were assessed. The significance of differences was set at *p* ≤ 0.05 and *p* ≤ 0.01. Pearson’s coefficients of correlation were also calculated between the analysed traits.

## 3. Results

Significant differences (*p* ≤ 0.01) were found between hen strains for all interior egg and shell quality traits included in Table 1, Table 2 and Table 3, with the exception of yolk colour lightness (L*) and redness (a*). Eggs collected at 7:00 h were characterised by lower total weight and albumen weight (by 1.7 g on average in both parameters analysed), higher albumen and yolk pH values (by 0.15 and 0.17 on average, respectively) and lower shell porosity (by 2.4 pores/cm^2^ on average) compared with those collected at 13:00 h (*p* ≤ 0.01). When evaluating the quality of eggs collected at different time intervals within each strain, it was found that eggs from R-11 hens laid in the morning (collected at 7:00 h) were characterised by lower weight and lower albumen content compared with those collected at 13:00 h. There were no significant differences (*p* > 0.05) regarding the remaining traits between eggs collected at different times of day. 

Irrespective of the time of egg collection, eggs of the H-22 hens were characterised by highest total weight (64.6 g at 7:00 h and 66.1 g at 13:00 h) and eggs of the native Polish strain Z-11 by the lowest (58.5 g at 7:00 h and 57.8 g at 13:00 h). Yellow-legged Partridge hens (Ż-33) laid eggs with the largest yolk weight (20.3 g at 7:00 h) and Green-legged Partridge hens (Z-11) laid eggs with the highest pH of yolk (6.45 at 7:00 h and 6.25 at 13:00 h) and albumen (8.16 at 13:00 h). Shells from the R-11 hens collected in the afternoon were characterised by the highest crushing strength (41.5 N) compared with shells from other strains.

Significant differences were noted in the lysozyme content and activity in thick and thin albumen in eggs laid by hens of different strains, while no effect of egg collection time on this trait was confirmed (*p* > 0.05) (Table 4). Irrespective of the time of egg collection, the highest levels of lysozyme in thick albumen were found in eggs from Ż-33 (0.174% at 7:00 h; 0.170% at 13:00 h) and Z-11 hens (0.172% at 7:00 h; 0.173% at 13:00 h), and the lowest—in R-11 (0.160% at 7:00 h and 0.153% at 13:00 h). In addition, the Ż-33 strain showed the highest values for the lysozyme parameters in both analysed albumen fractions of eggs collected at 7:00 h (0.174% and 36,978 U/mL—thick albumen; 0.312% and 66,448 U/mL—thin albumen). The lowest lysozyme activity in the thick albumen was recorded in eggs of the R-11 strain (34,003 U/mL—7:00 h; 32,444 U/mL—13:00 h) and in the thin albumen in eggs of the Z-11 strain (48,737 U/mL—7:00 h; 47,746 U/mL—13:00 h). 

As can be seen from Appendix A, irrespective of the egg collection time, the lysozyme content of egg albumen was significantly negatively correlated with egg and albumen weight, albumen height, Haugh units, yellowness (b*) and shell characteristics, such as weight, thickness, porosity and crushing strength. A significant positive relationship in this respect was found for albumen pH and shell yellowness (b*). Lysozyme activity in the thick albumen of eggs collected in the morning was correlated with traits such as air cell height (0.454*), yolk weight (0.372*), yolk pH (−0.486*) and redness (a* 0.295*). In contrast, no correlation (*p* > 0.05) was observed between any egg quality traits and lysozyme activity in the thick albumen of eggs collected at 7:00 h, but in eggs collected at 13:00 h, a significant negative correlation was confirmed for albumen height (−0.277*), Haugh units (−0.269*), redness (a* −0.301*) and crushing strength (−0.273*).

## 4. Discussion

The selection of the strains of hens, the eggs of which were investigated, was based on an analysis of the results of previous studies on factors shaping lysozyme levels in egg albumen. As the studies in [6,21] found increased levels of lysozyme in eggs from older hens, the subject of our study was eggs collected from hens at 56 weeks of age. Additionally, Ref. [5] found that lysozyme content in eggs is related to shell colour, so white eggs from H-22 hens, cream-coloured eggs from Ż-33 and Z-11 hens and brown-shelled eggs from R-11 hens were selected for the study.

The present study shows a large variation in interior egg and shell quality traits between hen strains, consistent with our previous research on eggs of native hen strains [5,6] and with the findings of [15]. The lower egg weight of Z-11, R-11 and Ż-33 hens, compared with H-22, is a characteristic of these old, native hen strains due to their genetic diversity and the abandonment of work on improving productivity and egg quality traits in conservation flocks [22]. A study by [23] shows that modern commercial hybrid hens lay eggs of uniform weight at 35–65 weeks of age, undoubtedly the result of breeding work on pedigree farms. The weight of eggs from hens of the strains included in the conservation programme was characterised by considerable variability (Table 1). Similar variation in egg weight of hens was obtained in earlier studies by [3,9]. A study by [24] found that the egg weights of native hen strains are significantly lower compared to commercial hybrids. Furthermore, in eggs from commercial hens, there is a high correlation between egg weight and albumen and yolk weight, which was not recorded among R-11, Ż-33 and Z-11 eggs in our study, as large yolks and varying albumen weights were recorded in low-weight eggs. However, the high percentage of yolks in the eggs studied was confirmed as in the studies in [3,24].

In our study, we observed lower egg weight and a smaller proportion of albumen in eggs laid in the morning than in the afternoon, and these results were the opposite of those obtained by [25] in a similar study on eggs from high-producing hens. On the other hand, in a later study by [1], there was a tendency—similar to our study—for the weight of eggs laid in the afternoon by commercial hybrids to decrease, but the opposite trend was noted in eggs from Moravia hens of the native breed. 

Eggs are considered fresh if the minimum value of Haugh units is 60, and in the eggs studied, this indicator, regardless of genotype and time of laying, was above 66. This means that eggs from all hens had good freshness results. However, the level of albumen height and Haugh units in eggs from Z-11 and Ż-33 hens was alarmingly low. Studies show that in eggs from older layers (as in our experiment), a lower value of Haugh units is observed, which, however, is >80 [3,23]. The time of laying did not significantly affect the freshness indices of the eggs, but a slightly higher level of these parameters was recorded in eggs laid in the afternoon. The same trend was also noted by [25] in a similar study of eggs from ISA Brown commercial hybrids.

The greater value of the albumen and yolk pH index recorded in all eggs laid in the morning is indicative of an alkalinisation of the environment inside the egg and a change in the albumen structure, and the maintenance of its gel-like structure is determined by the existence of an interaction between lysozyme and ovomucin, which affects its functional characteristics [26]. Yolk pH values were at levels similar to those obtained by [27], and the pH in the whites of the tested eggs was at a lower level than in studies of commercial hybrid eggs [28]. 

According to [29], the relationship between egg pH and lysozyme activity is most stable at a pH of about 8.0, while [30] showed that the enzymatic stability of lysozyme activity is preserved at pH 5.3–6.0. In our study, the average pH of albumen was 7.99 and 7.84 for eggs laid in the morning and in the afternoon, respectively, and it was close to the results of similar studies of eggs from different breeds of hens by [29,31].

The genotype of the hens significantly influenced the differences recorded in the shell quality traits of the eggs studied, and the results obtained are consistent with the studies by [1,9]. The time of laying significantly influenced the shell porosity, which was significantly higher in eggs laid in the afternoon than in the morning, which was probably the reason for the reduced crushing strength of the shells. In a study by [1], the effect of genotype and egg laying time on shell strength was found for hens of the native Moravia breed kept on litter. Other research shows that eggshell thickness and eggshell Ca content of eggs from rye-fed hens can be improved by xylanase supplementation [32]. 

Since eggshell colour is a genetically determined trait, it did not change due to the time of laying. In contrast, studies by [2,13] show that in native and hens for backyard flocks, there is a relationship between shell pigmentation and the concentration of lysozyme and ovotransferrin in eggs. A similar relationship was observed in our earlier study [5], in which white-shelled eggs (H-22) contained the least lysozyme and those with cream and light brown shells (Z-11, Ż-33 and R-11) contained the most. The results obtained in our study did not provide a conclusive answer for the eggs of H-22, Z-11 and R-11 hens. Significant differences in lysozyme content and activity between hen strains are apparent from the presented study, with the highest lysozyme content and activity found in the eggs of Ż-33 hens, which is consistent with the results of our previous studies [5,6]. In contrast, the lowest levels and activity of lysozyme were found in the thin albumen of Z-11 eggs, which was also confirmed by the results of a study of eggs from this strain of hens in [29]. 

There was a positive correlation between lysozyme content in egg albumen and shell colour, but only for yellowness (b*) and redness (a*). In contrast, no differences (*p* > 0.05) were confirmed in the lysozyme content and activity in egg albumen depending on the time of laying. The authors of [33], who examined eggs of seven strains from a domestic breeding farm, found significant negative correlations between albumen pH and lysozyme levels in eggs only in two strains, while in our study, a positive significant correlation was noted between albumen pH and its content and activity in albumen, but only in eggs laid in the afternoon. As noted in [13], the influence of genotype and environmental factors and their mechanism of action on the deposition of antimicrobial compounds in egg albumen is not fully understood, and published research results in this area, such as those presented in this work, remain inconclusive. In the study presented here, a significant negative relationship between egg weight and lysozyme content in the albumen was found, as in the work of [28], but this relationship was not reported for lysozyme activity. However, a significant negative relationship was confirmed between Haugh units and lysozyme content and activity in both thick and thin albumen, which is consistent with the results of [21] and with studies on eggs from native hen breeds [6]. Negative correlations were observed between egg shell weight, thickness, porosity and strength and lysozyme content in thin and thick albumen irrespective of egg laying time, while no such correlations were statistically confirmed in the study in [5].

Based on our previous and current research results [5,6], we can conclude that the quality of eggs as well as the level and activity of lysozyme depend more on the genotype and age of the hens than on other factors, including laying eggs.

## 5. Conclusions

Our work studied the potential effect of egg laying time on certain egg traits, including lysozyme content. We demonstrated that eggs collected in the morning were characterised by lower total weight and albumen weight, as well as a smaller shell pore number, whereas albumen and yolk pH were higher than eggs collected in the afternoon. The time of laying did not affect the lysozyme content and activity in the albumen. It is worth noting that eggshell traits, albumen height and Haugh unit showed a significant negative correlation with lysozyme content and activity in the albumen. Finally, based on our results, genotype influenced the studied egg quality traits, lysozyme content and activity more than the egg laying time.

## Figures and Tables

**Table 1 animals-13-01611-t001:** Effect of oviposition time and hen genotype on egg weight and albumen quality traits.

Time of Egg Collection (h)	Strain Symbol	Egg Weight (g)	Albumen Weight (g)	Percentage of Egg Albumen (%)	Albumen Height (mm)	Haugh Unit	Albumen pH
7:00	H-22	64.6	39.3	60.7	7.12 ^a^	82.9	7.80 ^a^
	R-11	58.8 ^a^	24.6 ^a^	58.8 ^a^	6.61	81.9	7.90 ^a^
	Ż-33	60.0	34.3	57.2	4.78	67.9	8.21 ^a^
	Z-11	58.5	35.6	60.8	5.26 ^a^	68.9 ^a^	8.05
Mean		60.5 ^x^	36.0 ^x^	59.4	5.94	75.4	7.99 ^x^
13:00	H-22	66.1	40.2	60.7	7.72 ^b^	86.2	7.62 ^b^
	R-11	63.9 ^b^	40.4 ^b^	63.0 ^b^	7.07	82.8	7.67 ^b^
	Ż-33	60.9	35.7	58.6	5.13	69.4	7.89 ^b^
	Z-11	57.8	24.7	60.1	4.71 ^b^	66.7 ^b^	8.16
Mean		62.2 ^y^	37.7 ^y^	60.6	6.16	76.3	7.84 ^y^
SEM		0.473	0.455	0.439	0.120	0.852	0.024
Significance of differences (*p* value)							
Collection time		0.04	0.03	0.16	0.09	0.38	0.00
Genotype		0.00	0.00	0.05	0.00	0.00	0.00
Genotype × collection time		0.07	0.03	0.18	0.01	0.02	0.00

Notes: Strains of hens: Green-legged Partridge (Z-11), Yellow-legged Partridge (Ż-33), Rhode Island Red (R-11), Leghorn (H-22); Columns with different superscripts (^x,y^) are significantly different (*p* < 0.05) between egg collection time for mean values of all strains; Columns with different superscripts (^a,b^) are significantly different (*p* < 0.05) between egg collection time, separately for each strain; SEM—standard error of the means.

**Table 2 animals-13-01611-t002:** Quality traits of eggs laid in the morning and in the afternoon.

Time of Egg Collection (h)	Strain Symbol	Yolk Weight (g)	Percentage of Yolk Weight (%)	Yolk Colour			Yolk pH
				L*	a*	b*	
7:00	H-22	19.2	29.8	51.3	16.7	39.7	6.06 ^a^
	R-11	18.8	32.1	49.4	16.3	32.4	6.25 ^a^
	Ż-33	20.3	33.7	49.3	18.2	32.5	6.16 ^a^
	Z-11	17.8	30.4	51.0	15.7	32.7	6.45 ^a^
Mean		19.0	31.5	50.3	16.7	34.3	6.23 ^x^
13:00	H-22	19.9	30.2	49.9	18.0	39.7	5.93 ^b^
	R-11	19.8	31.0	48.8	16.9	32.2	6.08 ^b^
	Ż-33	19.7	32.5	49.3	16.2	26.7	5.98 ^b^
	Z-11	17.9	31.0	49.2	15.4	28.2	6.25 ^b^
Mean		19.3	31.2	49.3	16.6	31.7	6.06 ^y^
SEM		0.153	0.205	0.279	0.297	0.959	0.017
Significance of differences (*p* value)							
Collection time		0.22	0.36	0.08	0.88	0.14	0.00
Genotype		0.00	0.00	0.19	0.13	0.00	0.00
Genotype × collection time		0.20	0.12	0.66	0.20	0.57	0.63

Strains of hens: Green-legged Partridge (Z-11), Yellow-legged Partridge (Ż-33), Rhode Island Red (R-11), Leghorn (H-22); Columns with the different superscripts (^x,y^) are significantly different (*p* < 0.05) between egg collection time for mean values of all strains; Columns with different superscripts (^a,b^) are significantly different (*p* < 0.05) between egg collection time, separately for each strain; SEM—standard error of the means. L*—lightness, a*—redness, b*—yellowness.

**Table 3 animals-13-01611-t003:** Shell quality of eggs laid in the morning and in the afternoon.

Time of Egg Collection (h)	Strain Symbol	Shell Weight (g)	Shell Thickness (µm)	Shell Colour			Shell Porosity (pores/cm^2^)	Shell Crushing Strength (N)
				L*	a*	b*		
7:00	H-22	6.11	363	92.6	0.11	4.14	44.3	40.4
	R-11	5.34	340	75.0	7.64 ^a^	19.5	44.1	38.2
	Ż-33	5.41	332	83.4	3.17	16.0	34.9	36.6
	Z-11	5.09	322	86.4	0.63	13.8	40.0	31.9
Mean		5.49	339	84.4	2.58	13.4	40.8 ^x^	36.8
13:00	H-22	5.98	352	93.0	0.99	3.03	47.1	36.3
	R-11	5.72	340	74.9	8.84 ^b^	19.9	46.1	41.5
	Ż-33	5.43	330	83.4	3.44	15.4	37.8	33.2
	Z-11	5.16	328	87.4	0.65	12.8	42.0	33.5
Mean		5.57	338	84.7	2.98	12.8	43.2 ^y^	36.1
SEM		0.058	2.64	0.633	0.350	0.580	0.510	0.839
Significance of differences (*p* value)								
Collection time		0.42	0.72	0.60	0.15	0.18	0.00	0.71
Genotype		0.00	0.00	0.00	0.00	0.00	0.00	0.00
Genotype × collection time		0.36	0.65	0.81	0.41	0.62	0.96	0.29

Notes: Strains of hens: Green-legged Partridge (Z-11), Yellow-legged Partridge (Ż-33), Rhode Island Red (R-11), Leghorn (H-22); Columns with the different superscripts (^x,y^) are significantly different (*p* < 0.05) between egg collection time for mean values of all strains; Columns with different superscripts (^a,b^) are significantly different (*p* < 0.05) between egg collection time, separately for each strain; SEM—standard error of the means; L*—lightness, a*—redness, b*—yellowness.

**Table 4 animals-13-01611-t004:** Effect of oviposition time and hen genotype on lysozyme content and activity in egg albumen.

Time of Egg Collection (h)	Strain Symbol	Lysozyme Content (%)		Lysozyme Activity (U/mL)	
		Thick Albumen	Thin Albumen	Thick Albumen	Thin Albumen
7:00	H-22	0.171	0.303	36,412	64,323
	R-11	0.160	0.311	34,003	66,164
	Ż-33	0.174	0.312	36,978	66,448
	Z-11	0.172	0.289	36,270	48,737
Mean		0.169	0.289	35,916	61,418
13:00	H-22	0.162	0.307	34,428	65,314
	R-11	0.153	0.303	32,444	64,464
	Ż-33	0.170	0.314	36,128	66,731
	Z-11	0.173	0.225	36,695	47,746
Mean		0.164	0.287	34,924	61,064
SEM		0.002	0.004	391	854
Significance of differences (*p* value)					
Collection time		0.318	0.742	0.191	0.732
Genotype		0.006	0.000	0.007	0.000
Genotype × collection time		0.395	0.787	0.691	0.791

Notes: Strains of hens: Green-legged Partridge (Z-11), Yellow-legged Partridge (Ż-33), Rhode Island Red (R-11), Leghorn (H-22). Significance at *p* < 0.05.

## Data Availability

Not applicable.

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
