# Peer review of "Effect of Hen Genotype and Laying Time on Egg Quality and Albumen Lysozyme Content and Activity"

_animals, 2023, doi:10.3390/ani13101611_

Round 1
Reviewer 1 Report
The manuscript is very well written and shows interesting results about the biodiversity of hens of the four breeds included in the conservation program in Poland.
I suggest publication of the manuscript.
Author Response
We want to thank Reviewer for the positive opinion about the manuscript.

Reviewer 2 Report
The manuscript is interesting but has to be largely improved to be considered. My comments aim to increase the scientific soundness and clarity of it.
Line 21 – a brief introductory sentence is missing.
Line 35,70, 324, 327, 350 – In some places of the manuscript double coma is used.
Line 74 – please formulate your hypothesis correctly.
Line 61 – the authors did not mention that also supplements like xylanse substantially improves bone strength and eggshell quality (see recent work of MuszyÅ„ski et al. Assessing Bone Health Status and Eggshell Quality of Laying Hens at the End of a Production Cycle in Response to Inclusion of a Hybrid Rye to a Wheat–Corn Diet. Vet. Sci. 2022, 9, 683. https://doi.org/10.3390/vetsci9120683). This issue is definitely in line with the topic of the current article and in Reviewer’s opinion the authors should briefly address this issue in the Introduction as well as acknowledge this work.
Line 84 – ad libitum should be written in italics.
Line 77 – The whole methodology chapter is poorly written. It is too enigmatic and many important details are missing. I consider it necessary to develop a more precise description of the methods used. The authors should remember that a reader should be able to repeat their experiment.
Line 148, 149 – what “…” stands for? Only two sets of letters A and B (or a and b) are used. It is unclear, significance was marked with letters or asterisks?
Line 309 – The authors apparently did not understand how the correct conclusion should be formulated. Some conclusions (first) are too speculative and directly not supported by the results. Please rearrange it.
Author Response
Dear Reviewer,
Thank you for your precious comments. We improved the manuscript according to them and hope it is more adequate now.
Regards

Reviewer 3 Report
Dear Authors,
Is there any significant difference between your current study and the other study below?
“Józefa Krawczyk1, Lidia Lewko , JolantaCalik. 2021. Effect of laying hen genotype, age and some interior egg quality traits on lysozyme content. Ann. Anim. Sci., Vol. 21, No. 3 (2021) 1119–1132”
Best regards,
INTRODUCTION
It is thought that the introduction part is missing. It should be given information about the local genotypes and their importance in terms of the subject.
It is seen that the reference style does not match the writing style of the journal. This makes it difficult to analyze the references.
The sentence specified in the introduction, "Polish native breeds of hens," needs to be rewritten. The Rhode Island Red and Leghorn breeds are expressed as a Polish breed.
Line 41: Too many references are given.
Lines 48-49: These references are related to the topic you mentioned. “The lysozyme in egg albumen provides natural protection against infection of the embryo until the embryo produces immunoglobulins [10,11].”
Line 51: The reference is not fully written. “10] also showed a positive effect of lysozyme on the hatchability of Japanese quails.” Also, this reference is not about hatchability in quail.
Line 86: Replace “life” with “age”
MATERIALS AND METHODS
Choosing only two times will adversely affect the correct results of the experiment. Why didn't you choose intermittent times (like 6–8 a.m. and 13–15 p.m.)?
Eggs laid between 7 and 8 o'clock are considered among the eggs laid in the afternoon.
Lines84-85: It would be appropriate to give the energy values of the feed.
The hen-day egg production of the hens should be specified. It should be stated whether the eggs are taken randomly or whether the cracked, dirty eggs are eliminated.
It was stated that a total of 336 eggs were used in the study. Check the egg numbers (28 eggs x 2 (times) x 4 (breeds) =224 eggs.)
You must specify how it measures the eggshell color. How the oval shape of the egg was measured with a flat objective.
You have to provide the method of measurement of the albumen height, Haugh unit, albumen, and yolk pH.
Statistical analyses
There are only two hypotheses in statistics, they are H0 and H1. In the null hypothesis (H0), the distribution is gaussian, the treatment effect is not significant. In the H1 hypothesis, the situation is the opposite. In the natural, health, and engineering sciences, the acceptance probability of the null hypothesis spreads up to 0.95/1. If a P value of 0.049 is found in a study with a significance level of 0.05, H0 is rejected. But this does not mean that the "more significant" difference occurs when a P value of 0.01 or 0.001 is found in the same study. There is no H2 or H3 hypotheses in statistics science. Therefore, use only one of these levels (0.05, 0.01, 0.001) in your studies.
RESULTS
It is very difficult to analyze the tables. For example, although a sentence like "lines 175-181, and lines 185-193" is used, the statistical difference between genotypes is not given in superscript. Therefore, the figures for the main effects should be indicated in the table.
If you make the tables below, the importance of the main effects (time and genotype) will be shown more effectively.
|
treatments |
Egg weight |
|
|
times |
|
|
|
7 |
|
|
|
13 |
|
|
|
sem |
|
|
|
P-value |
|
|
|
genotype |
|
|
|
x |
|
|
|
y |
|
|
|
z |
|
|
|
t |
|
|
|
sem |
|
|
|
P-value |
|
|
|
interaksiyon |
|
|
|
7X x |
|
|
|
7 x y |
|
|
|
7 x z |
|
|
|
… |
|
|
|
… |
|
|
|
…. |
|
|
|
sem |
|
|
|
P-value |
|
Table 1 and 5: Replace “haugh units” with “haugh unit”
Line 144: This sentence is incompatible with academic writing. Please use like this: "no significant differences."
Use the term either genotype or strain throughout the article.
Line 202: Replace “Haugh units” with “Haugh unit”
Line 207: Delete one of the "p>0.05"
Check the shell thickness unit (Table 3, and Table 5)
DISCUSSION
Line 270-271: You said that “The differences recorded in the shell quality traits of the eggs studied were significantly influenced by the genotype of the hens”. When the table is examined, it cannot be determined which genotypes have differences. Therefore, the display of the data in the table should be rearranged.
Line 278: “fancy hen” ?
CONCLUSIONS
Lines 314-316: Please check the sentences.
Lines 319-320: Please check the sentences.
The authors should explain that what is differences between this manuscript and the articles below.
Author Response
Dear Reviewer,
Thank you for your precious comments. We improved the manuscript according to them and hope it is more adequate now.
Regards.

Round 2
Reviewer 2 Report
Some of my recommendations were considered but most of majors’ points of criticism not. Introduction and abstract improvements as well as changes in conclusions are not included in the revised version.
1. Still introductory sentence in abstract is missing.
2. The authors studied the eggshell traits and any works in this field are relevant.
3. Tables are not fully corrected (I still do not know how to correctly read a* b* and L* significant values. They are significant in relation to what?)
4. The conclusions should be concise without any points and data presentation. First point is repetitive description of the results. Also some future perspectives should be added.
Author Response
Dear Reviewer,
Please find our response to Review Report in the attachment.
Yours sincerely.
Józefa Krawczyk

Reviewer 3 Report
Dear Authors,
I think the corrections made are not enough. It is a big mistake to give the wrong albumen height and shell thickness, especially. I think this work is a repetition of your own work. The line numbers of the revisions are shown incorrectly. It is clear that not enough attention has been done to the revision.
Best regards.
Author Response

(The authors gave the same response as above.)
